# Influence of Metabolic Syndrome on Cancer Risk in HBV Carriers: A Nationwide Population Based Study Using the National Health Insurance Service Database

**DOI:** 10.3390/jcm10112401

**Published:** 2021-05-29

**Authors:** Jung Wan Choe, Jong Jin Hyun, Bongseong Kim, Kyung-Do Han

**Affiliations:** 1Department of Internal Medicine, Korea University Ansan Hospital, Ansan 15355, Korea; jwchoe@korea.ac.kr; 2Department of Biostatistics, College of Medicine, The Catholic University of Korea, Seoul 06591, Korea; qhdtjd12@gmail.com

**Keywords:** metabolic syndrome, hepatitis B virus, neoplasms

## Abstract

Purpose: Hepatitis B virus (HBV) infection and metabolic syndrome (MS) are known independent risk factors for hepatocellular carcinoma (HCC) and other extrahepatic organ malignancies. The purpose of this study was to investigate whether MS and HBV have synergistic effects on cancers and to examine whether increasing the number of MS components could lead to higher risk of cancer development. Materials and Methods: We evaluated data from 1,504,880 HBV-infected adults who underwent a regular HCC screening program provided by the Korean National Health Insurance Service between 2009 and 2016. Results: The prevalence of MS in Korean HBV patients was 38.7% (582,449/1,504,880). Among individuals with HBV infection, the presence of MS was associated with an increased risk for the majority of malignancies except for HCC (HR = 0.862, *p*-value < 0.05). The presence of a higher number of MS components was associated with a significantly increased risk of developing cancers in most organs; only HCC was negatively associated with an increasing number of MS components (*p* < 0.01). Conclusions: Our data show that the presence of MS increases the risk for most malignancies, excluding HCC. Moreover, we found that as the number of MS components increased, the risk for most cancers also increased; this trend was reversed in HCC.

## 1. Introduction

Hepatitis B virus (HBV) infection is known to play a role in chronic liver damage and the subsequent development of hepatocellular carcinoma (HCC) and cholangiocarcinoma [1,2]. While HBV is generally considered a hepatotropic virus, various studies have reported that HBV can exist in extrahepatic organs such as the pancreas, kidneys, skin, gastric mucosa, lymph nodes, spleen, bone marrow, colon, and testes [3]. Additionally, chronic HBV infection has been shown to increase the risk of extra-hepatic malignancies such as pancreatic cancer, gastric cancer, non-Hodgkin lymphoma and colon cancer [4].

Metabolic syndrome (MS), a condition that includes central obesity, hypertension, dyslipidemia, and diabetes, is a growing public health concern due to its high prevalence and poor long-term outcomes. While MS was initially only considered a risk factor for diabetes and cardiovascular disease, recent studies have suggested that it is also a risk factor for various types of cancer [5]. MS is known to be associated with a higher risk of liver, colorectal, and bladder cancers in men, and endometrial, pancreatic, colorectal, ovarian, and postmenopausal breast cancers in women. The pathophysiology that underlies the association of MS with cancer remains poorly understood to date. Moreover, whether MS as a whole carries a higher risk than each separate MS component is also unclear.

Despite the fact that there is a dearth of information about malignancies developing in extrahepatic sites as well as information describing which components of MS have synergistic effects on carcinogenesis, few studies have investigated the close association between the components of MS and HBV-related liver cirrhosis and HCC [6,7].

Therefore, the main objective of this study was to elucidate the synergistic association between MS and HBV in HCC and extrahepatic organ malignancies, and to investigate which MS components are associated with increased cancer risk.

## 2. Materials

### 2.1. Study Population

The National Health Insurance Service (NHIS) database, which is managed by the Korean government and covers approximately 97% of the Korean population, was used for this study [8]. Between 1 January 2009, and 31 December 2016, data on 2,567,812 HBV-infected adults over 40 years old were collected from the Korean National Liver Cancer Surveillance Program provided by NHIS. International Statistical Classification of Diseases and Related Health Problems, Tenth Revision (ICD-10) codes B16, B16.1, B16.2, B16.9, B17.0, B18.0 or B18.1 were used to select patients with HBV. Patients who received a diagnosis of HCC or other malignancies within 1 year of enrollment or had a history of malignancy at any time were excluded. Patients who failed to attend follow-up appointments and individuals who had missing data were also excluded. Finally, a total of 1,504,880 patients were included for this study (Figure 1). Presence of HCC was identified by ICD-10 code C22, and non-HCC cancers were also identified using ICD-10 codes as shown in Appendix A.

This study was approved by the Institutional Review Board of Korea University Ansan Hospital (No. 2019AS0012) and the research was conducted in accordance with the Helsinki declaration.

### 2.2. Measurement of Variables

During a Korean national health examination, waist circumference, body mass index, and systolic and diastolic pressure are typically measured. In addition, the levels of fasting plasma glucose, triglycerides (TG), total cholesterol, and high-density lipoprotein cholesterol (HDL-c) are also obtained. Lifestyle factors, such as smoking status, alcohol consumption, and physical activity, are also determined during the examination via a self-reported questionnaire. Smoking status is categorized as current, former, or never smokers. Alcohol intake is assessed by the frequency of ingestion and average amount of alcohol consumed each time. (non, mild (mean consumption < 30 g/day), or heavy (mean consumption ≥ 30 g/day)), Physical activity is determined by the number of times each week the subject exercises; exercising less than three times per week is defined as lacking physical activity. 

### 2.3. Criteria of Metabolic Syndrome

MS was defined as the presence of at least three of the following five criteria proposed by the American Heart Association and the National Heart, Lung and Blood Institute, together with the International Diabetes Federation, in 2009.

Central obesity (waist circumference >94 cm for males and >80 cm for females), according to the Asian-Pacific criteriaRaised TG ≥ 150 mg/dL (1.7 mmol/L) or specific treatment for this lipid abnormalityReduced HDL-c < 40 mg/dL (1.03 mmol/L) in males and <50 mg/dL (1.29 mmol/L) in females or specific treatment for this lipid abnormalityRaised blood pressure, systolic ≥ 130 or diastolic ≥ 85 mm Hg, or treatment of previously diagnosed hypertensionRaised fasting plasma glucose ≥ 100 mg/dL (5.6 mmol/L), or previously diagnosed type 2 diabetes.

### 2.4. Statistical Analysis

SAS System version 9.4 for Windows was used for all statistical analyses. All categorical variables are expressed as percentages and all continuous variables are expressed as mean ± standard deviation. Multivariate Cox regression analysis was performed to examine the hazard ratio (HR) and confidence interval for the relationship between MS status and malignant disease. The multivariable analyses were adjusted for factors including age, sex, body mass index, smoking status, alcohol consumption and physical activity, *p* values < 0.05 were considered statistically significant.

## 3. Results

The prevalence of MS in Korean HBV patients was 38.7% (582,449/1,504,880).Table 1 shows baseline characteristics of the study population. The median age of the overall study population was 53 years. Patients with MS tended to be older and obese, consumed significantly more alcohol, and had a higher proportion of MS components. Among individuals infected with HBV, the presence of MS was associated with an increased risk of developing malignancies other than HCC (Figure 2, Table 2). As for HCC, the relative risk of incidence rate was 0.862, which was remarkably lower in HBV carriers with MS than those without (3.57 vs. 4.02, *p* < 0.0001). The cumulative incidence of overall malignancy was significantly higher in HBV carriers with MS than in those without MS (14.21 vs. 12.20, *p* <0.001), however, this trend was reversed for HCC (Figure 3).

Table 3 shows the relative malignant neoplasm risk associated with five metabolic factors. Among the components of the MS, high serum TG and low HDL-c levels were negative contributing factors for HCC. On the other hand, the two aforementioned components were independent risk factors for malignancy in other organs. Additionally, increased waist circumference, high blood pressure, and high blood glucose level in HBV patients were positive contributing factors for all types of cancers including HCC (Table 3).

The relationship between malignant neoplasms and the number of individual MS components is shown in Table 4. The presence of a higher number of MS components was associated with a significantly increased risk of developing cancers in most organs; only HCC was negatively associated with an increased number of MS components.

## 4. Discussion

In this study, we investigated the relationship between MS and malignancy, and determined the effect of individual MS components on malignancy in patients with HBV infection. We found that MS in HBV-infected individuals was associated with an increased risk of most malignancies, except for HCC. Additionally, most cancers showed an increase in prevalence as the number of MS components increased, however, this trend was reversed in HCC. Interestingly, high serum TG and low HDL-c levels were inversely associated with HCC risk.

HBV is a well-known cause of HCC. The integration of HBV DNA fragments into chromosomal DNA can result in the modulation of gene expression, especially of oncogenes and tumor suppressor genes in the liver. In recent studies, malignancies other than HCC have been shown to be associated with HBV infection [9,10]. In fact, HBV has been detected in several types of extrahepatic organ tissues, suggesting a potential role for HBV in the tumorigenesis of non-liver cancers [11]. Several studies have suggested that the X protein from HBV can bind and interfere with the components of DNA repair machinery and p53 in response to DNA damage, thereby increasing the risk of non-liver cancers [12,13].

It is generally accepted that metabolic disorders are closely associated with the occurrence and development of malignancies. However, there are insufficient data about the epidemiology of MS and the synergistic influence of MS on HCC and extrahepatic cancer development in patients with HBV. Some recent research has aimed to determine if the effect of individual components of MS were stronger than the overall effect of MS as a whole. For instance, in the case of endometrial cancer, obesity on its own carries a higher risk than MS as a whole [14]. As another example, while MS has been shown to be slightly associated with prostate cancer, a stronger association exists between prostate cancer and hypertension [15]. Therefore, assessment of individual MS components may help future therapies target the important factors of MS that contribute most to the development of malignancies. In the current study, MS tended to increase nearly all cancer risks in individuals with HBV infection, similar to the trend seen in the general population between MS and cancer more generally [16,17]. A higher number of MS components was associated with a significantly increased risk of all cancers types. However, our data show that HCC was inversely related to the presence of MS and the number of MS components. Interestingly, a low HDL-c and high level of TG were negatively correlated with the risk of HCC in the current study.

With regard to HCC, an association between MS and HCC has previously been demonstrated independent of the HBV state. The progression of hepatic fibrosis in relation to MS could result from the direct stimulation of liver stellate cells by hyperinsulinemia and hyperglycemia, resulting in an increased production of connective tissue growth factors and a subsequent accumulation of extracellular matrix. These processes may increase hepatic injury from other common factors, including HBV.

The role of lipid metabolism in cancer development has also not been fully explored to date. Although the exact mechanism on how HBV-related diseases are closely associated with lipid metabolisms has not been clarified until now, there are several other studies supporting our findings which show an inverse relationship between hypertriglyceridemia and the risk of HCC among those with HBV infections [18,19,20]. Lipid and lipoprotein metabolism are regulated by cytokines, and malignant hepatocyte cells are known to produce large amounts of pro-inflammatory cytokines. For instance, interleukin (IL)-6, tumor necrosis factor, and IL-1 may all inhibit TG synthesis [21]. The loss of the negative feedback mechanism for cholesterol regulation as well as an increase in cholesterol synthesis by undifferentiated HCC cells may be responsible for the hypercholesterolemia seen in cirrhotic patients with HCC [22].

In an experimental HBV X (HBx) transgenic mouse model, lipid metabolomics were analyzed during the progression of HBx transgenic HCC [23]. Serum TG was mildly increased during the early stage, but was decreased in mice with HCCs. The serum levels of cholesterol was mildly increased during the early stage and significantly increased during the HCC stage. The study by Frank et al. also identified that the dysregulation of lipid genes by HBV generates a series of molecular alterations which collectively activates oncogenes or inhibits tumor suppressors, thereby leading to tumorigenesis [23]. This study supports our data that the aberrant lipid metabolism with serum low TG and high cholesterol in individuals with HBV might be associated with HCC development.

Unfortunately, our study could not preclude the possibility that low TG was a consequence of the change in general nutritional status during disease progression and that high HDL-c was the result of statin use. Hence, a prospective study is needed to elucidate the causative and consequent effects of the abnormal lipid profile seen in HBV-related HCC.

Recently, a study on the association of metabolic risk factors with risks of cancer and all-cause mortality in patients with chronic hepatitis B was published using a similar but different dataset from the NHIS database. In the study by Lee et al. [17], the risk of developing HCC and non-HCC cancers both increased in patients with MS, which was different from the result of our study that showed negative association of HCC with the presence of MS. There could be several reasons for this discrepancy. First, while we analyzed the lipid profile after subdividing it into TG and HDL-c, Lee et al. used only total serum cholesterol level. Second, the waist circumference was substituted for BMI as one of the components of MS in their study, whereas we adopted the classic definition of MS. Third, the average age of the patients was younger (46 (37–54), median (IQR)) compared to our study population (53.5 ± 9.77, mean ± SD), which could have limited the assessment of cancer development due to a comparatively younger study population. More importantly, the number of patients lost to follow-up was over 50% in their study which could have acted as a shortcoming and rendered the data less representative of the population of interest.

Interestingly, the study by Chen et al. [24] demonstrated that the relationship between MS and the occurrence of HCC was negligible. This is based on the observation in the HBV endemic area that the link between MS and HCC via the development of NASH was trivial compared to the predominant influence of HBV infection per se on HCC development. However, the impact of MS on HCC was also insignificant in the population without HBV infection in their study, thus not being able to demonstrate the widely accepted MS-NASH-HCC sequence.

Differently from the study by Chen et al., the result of another study from Taiwan was quite similar to our study [18]. In this study, the authors demonstrated that DM was the most important risk factor for developing HCC in HBV carriers, and that central obesity was responsible for a 33% increase in the risk of HCC. It was also shown that hypertriglyceridemia reduced the risk of HCC by 32%, which is in accordance with the finding of our study.

There are several limitations to our study. First, our study used retrospectively collected data from the NHIS database. Therefore, it is inevitable that there is missed information which may result in selection bias. However, this dataset comes from a nationwide population-based claims database with long-term follow-up of a large number of patients, thus reducing the potential for sampling bias. Second, since MS was determined by a single measurement at the time of the HCC screening program, changes in MS status over time were not available. Third, the status or activity of HBV infection and the usage of antiviral agents were not analyzed, which could have influenced the potential risk of HBV-related malignancy development. This is of particular importance in HCC as the associations between MS and liver disease can be influenced by the degree of underlying HBV-related liver pathology. In our dataset, there were no assessments of liver fibrosis either histologically, biochemically, or with transient elastography, which represents another significant limitation and provides a path for future study. Lastly, low TG and high HDL-c levels, which were associated with increased HCC risk, could have been attenuated by the presence of statin therapy. However, the Korean National Liver Cancer Surveillance Program database does not contain detailed information about self-administered medications.

In conclusion, the findings of this large population-based study indicate that MS in HBV carriers is associated with synergistic effect on the development of most malignancies, except for HCC. Moreover, the number of comorbid MS components was a risk factor for most cancers, but this trend was reversed in HCC. Finally, high serum TG and low HDL-c levels were inversely associated with HCC risk. This study could translate into the development of therapeutic strategies for MS by aiming to reduce cancer risk in patients with HBV infection. To date, the interactions between individual MS factors and HBV has yet to be fully elucidated. Pathophysiological studies are needed to better explore the possible biological mechanisms involved in the observed association between HBV infection and MS factors with regard to cancer development.

## Figures and Tables

**Figure 1 jcm-10-02401-f001:**
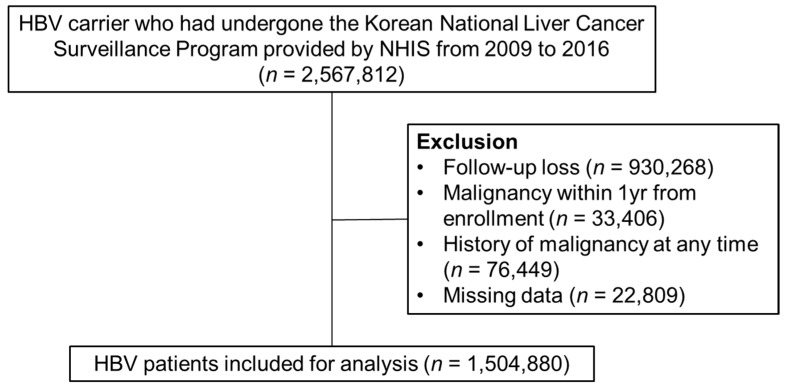
Flow chart of HBV-infected patients in the Korean NHIS database between 2009 and 2016. HBV, Hepatitis B virus; NHIS, National Health Insurance Service.

**Figure 2 jcm-10-02401-f002:**
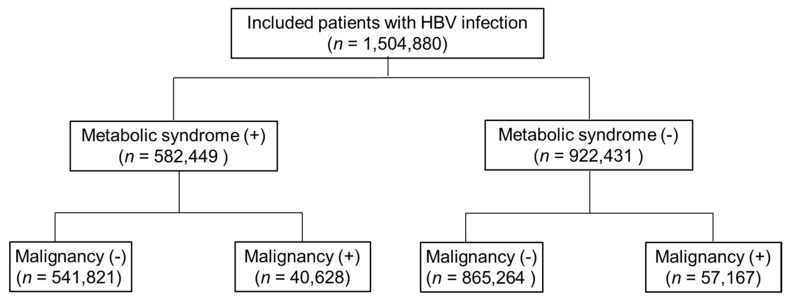
Diagram for incidences of metabolic syndrome and malignancy in HBV-infected patients. HBV, Hepatitis B virus.

**Figure 3 jcm-10-02401-f003:**
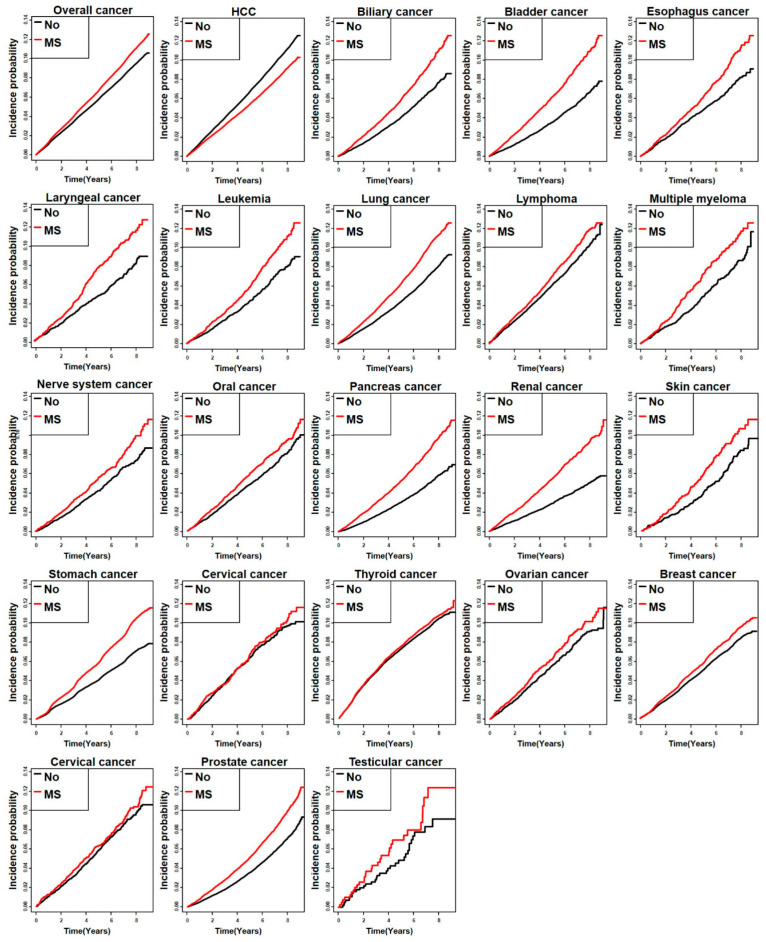
Cumulative incidence of cancer according to metabolic syndrome status in HBV-infected patients. MS, Metabolic syndrome; HCC, hepatocellular carcinoma.

**Table 1 jcm-10-02401-t001:** Characteristics of the study population.

Characteristics	Overall (*n* = 1,504,880)	Metabolic Syndrome (+) (*n* = 582,449)	Metabolic Syndrome (−) (*n* = 922,431)
Age, mean, years	53.5 ± 9.77	56.4 ± 9.88	51.55 ± 9.52
Sex			
Men	846,687 (56.3)	496,146(53.79)	350,541(60.18)
Smoke			
Non/Ex/Current	57.7%/18.9%/23.4%	54.2%/21.3%/24.5%	59.8%/17.5%/22.7%
Alcohol			
Non/Mild/Heavy	55.9%/33.9%/10.2%	54.7%/32.6%/12.7%	56.6%/34.8%/8.6%
Physical activity	320,665 (21.3)	122,217 (21.0)	198,448 (21.5)
BMI	24.2 ± 3.01	25.8 ± 3.28	23.32 ± 2.83
Metabolic syndrome factors			
Waist circumference > 94 cm for males and >80 cm for females	405,984 (27.0)	176,482 (30.3)	229,502 (24.9)
Systolic ≥ 130 or diastolic ≥ 85 mm Hg or diagnosed HTN	595,191 (39.6)	388,646 (66.7)	206,545 (22.4)
Fasting plasma glucose ≥ 100 mg/dL or type 2 DM	284,403 (18.9)	187,548 (32.2)	96,855 (10.5)
HDL-c < 40 mg/dL in males and <50 mg/dL in females	604,455 (40.2)	327,155 (56.2)	277,300 (30.1)
TG ≥ 150 mg/dL	645,159 (42.9)	315,442 (54.2)	329,717 (35.7)

*n* (%), BMI; body mass index, HTN; hypertension, TG, triglycerides; HDL-c, high-density lipoprotein cholesterol.

**Table 2 jcm-10-02401-t002:** Relationship between malignant neoplasms and the presence of metabolic syndrome components in HBV-infected patients.

Cancer Site	MS(+) (*n* = 582,449)		MS(−) (*n* = 922,431)			
	Case, No.	Incidence Rate No./1000 PYs	Case, No.	Incidence Rate No./1000 PYs	Adjusted HR	*p*-Value
**Overall cancer**	40,628	14.21	57,167	12.20	1.17 (1.15–1.18)	<0.0001
**HCC**	10,156	3.57	19,256	4.02	0.86 (0.84–0.88)	<0.0001
**Biliary**	1312	0.44	1489	0.31	1.45 (1.35–1.56)	<0.0001
**Stomach**	5022	1.71	5638	1.17	1.46 (1.41–1.52)	<0.0001
**Colorectum**	4260	1.45	4473	0.93	1.56 (1.50–1.63)	<0.0001
**Esophagus**	634	0.21	783	0.16	1.33 (1.20–1.48)	<0.0001
**Pancreas**	1468	0.50	1408	0.29	1.72 (1.60–1.85)	<0.0001
**lung**	4260	1.44	4937	1.02	1.42 (1.36–1.48)	<0.0001
**Renal**	1072	0.36	943	0.20	1.87 (1.71–2.04)	<0.0001
**thyroid**	2985	1.01	4744	0.98	1.03 (0.98–1.08)	0.29
**Laryngeal**	333	0.11	369	0.08	1.48 (1.28–1.71)	<0.0001
**Oral**	614	0.21	845	0.17	1.19 (1.07–1.32)	0.001
**Lymphoma**	826	0.28	1158	0.24	1.17 (1.07–1.28)	0.006
**Leukemia**	443	0.15	530	0.11	1.38 (1.21–1.56)	<0.0001
**Skin**	140	0.05	168	0.03	1.38 (1.10–1.72)	0.005
**Nerves**	432	0.15	559	0.12	1.27 (1.12–1.44)	0.0002
**Male**						
**Prostate**	2785	1.58	2911	1.12	1.43 (1.36–1.50)	<0.0001
**Testicular**	30	0.02	33	0.01	1.34 (0.82–2.20)	0.24
**Female**						
**Breast**	1717	1.45	2215	1.43	1.01 (0.95–1.08)	0.67
**Corpus**	286	0.24	489	0.22	1.10 (0.95–1.28)	0.19
**Ovarian**	254	0.21	458	0.21	1.04 (0.89–1.22)	0.59
**Cervical**	257	0.22	456	0.20	1.06 (0.91–1.23)	0.47

MS, Metabolic syndrome; HR, hazard ratio; HCC, hepatocellular carcinoma.

**Table 3 jcm-10-02401-t003:** Effect of each metabolic syndrome component on malignant neoplasm risk.

Risk Factors	Overall Cancer	Liver	Biliary	Bladder	Colorectal	Esophagus
Serum high TG ≥ 150 vs. <150	1.13 (0.99–1.28)	0.46 (0.45–0.47)	1.16 (1.07–1.25)	1.46 (1.34–1.59)	1.29 (1.24–1.35)	1.21 (1.09–1.34)
Serum low HDL cholesterol < 40 vs. ≥40	1.01 (1.00–1.02)	0.68 (0.66–0.70)	1.20 (1.11–1.29)	1.16 (1.07–1.26)	1.16 (1.11–1.21)	0.88 (0.80–1.01)
Serum abnormal glucose ≥ 100 vs. <100	1.31 (1.30–1.33)	1.53 (1.49–1.56)	1.52 (1.41–1.63)	1.72 (1.58–1.87)	1.53 (1.47–1.60)	1.80 (1.62–2.00)
Central obesity Yes vs. No	1.20 (1.18–1.21)	1.20 (1.18–1.23)	1.30 (1.20–1.41)	1.41 (1.29–1.54)	1.36 (1.31–1.43)	0.99 (0.88–1.11)
History of hypertension Yes vs. No	1.42 (1.41–1.44)	1.44 (1.40–1.47)	1.86 (1.72–2.01)	2.06 (1.88–2.27)	1.76 (1.68–1.84)	1.98 (1.76–2.21)
	**Laryngeal**	**Leukemia**	**Lung**	**Lymphoma**	**Multiple** **myeloma**	**Nerves**
Serum high TG ≥ 150 vs. <150	1.44 (1.24–1.67)	1.12 (0.99–1.27)	1.25 (1.20–1.31)	0.95 (0.87–1.04)	1.03 (0.88–1.21)	1.14 (1.01–1.29)
Serum low HDL cholesterol < 40 vs. ≥40	0.91 (0.78–1.06)	1.28 (1.13–1.45)	1.11 (1.07–1.16)	1.12 (1.02–1.22)	1.37 (1.17–1.60)	1.13 (0.99–1.28)
Serum abnormal glucose ≥ 100 vs. <100	1.86 (1.60–2.16)	1.19 (1.05–1.34)	1.48 (1.42–1.54)	1.21 (1.11–1.32)	1.19 (1.01–1.39)	1.13 (0.99–1.28)
Central obesity Yes vs. No	0.98 (0.83–1.15)	1.21 (1.06–1.39)	1.10 (1.05–1.15)	1.23 (1.12–1.35)	1.44 (1.22–1.69)	1.22 (1.06–1.39)
History of hypertension Yes vs. No	1.93 (1.64–2.27)	1.31 (1.15–1.49)	1.75 (1.67–1.83)	1.28 (1.17–1.40)	1.68 (1.42–1.98)	1.44 (1.27–1.64)
	**Oral**	**Pancreas**	**Renal**	**Skin**	**Stomach**	**Thyroid**
Serum high TG ≥ 150 vs. <150	1.12 (1.01–1.25)	1.29 (1.19–1.38)	1.44 (1.32–1.57)	1.16 (0.93–1.46)	1.19 (1.15–1.24)	1.03 (0.98–1.08)
Serum low HDL cholesterol < 40 vs. ≥40	0.96 (0.86–1.06)	1.33 (1.23–1.43)	1.31 (1.20–1.43)	1.18 (0.94–1.48)	1.05 (1.01–1.09)	1.17 (1.12–1.23)
Serum abnormal glucose ≥ 100 vs. <100	1.34 (1.21–1.48)	1.81 (1.68–1.95)	1.46 (1.34–1.59)	1.14 (0.91–1.42)	1.53 (1.47–1.59)	1.00 (0.96–1.05)
Central obesity Yes vs. No	0.97 (0.86–1.09)	1.33 (1.23–1.44)	1.57 (1.44–1.72)	1.33 (1.05–1.69)	1.28 (1.23–1.34)	1.08 (1.03–1.14)
History of hypertension Yes vs. No	1.51 (1.36–1.68)	1.83 (1.69–1.98)	2.24 (2.03–2.47)	1.66 (1.31–2.11)	1.62 (1.56–1.69)	0.92 (0.88–0.96)
	**Cervical**	**Ovarian**	**Corpus**	**Breast**	**Prostate**	**Testicular**
Serum high TG ≥ 150 vs. <150	0.88 (0.75–1.02)	1.04 (0.90–1.21)	0.95 (0.82–1.10)	1.04 (0.98–1.11)	1.11 (1.05–1.17)	1.36 (0.83–2.24)
Serum low HDL cholesterol < 40 vs. ≥40	0.92 (0.80–1.07)	0.96 (0.83–1.12)	1.01 (0.88–1.16)	1.03 (0.98–1.09)	1.35 (1.28–1.43)	0.99 (0.59–1.67)
Serum abnormal glucose ≥ 100 vs. <100	1.17 (1.01–1.37)	1.02 (0.88–1.20)	1.02 (0.88–1.18)	0.96 (0.91–1.02)	1.15 (1.09–1.21)	1.17 (0.71–1.91)
Central obesity Yes vs. No	1.31 (1.11–1.54)	1.20 (1.02–1.42)	1.43 (1.23–1.67)	1.00 (0.94–1.07)	1.26 (1.19–1.33)	1.82 (1.11–3.00)
History of hypertension Yes vs. No	1.12 (0.96–1.29)	0.99 (0.85–1.15)	1.06 (0.92–1.22)	1.06 (1.00–1.13)	1.90 (1.80–2.02)	1.19 (0.72–1.99)

TG, triglycerides; HDL, high-density lipoprotein.

**Table 4 jcm-10-02401-t004:** Relationship between malignant neoplasms and the number of individual metabolic syndrome components.

No. of MS Component	No. of Patients	Event	Duration	Rate	HR (95% CI)	*p*-Value for Trend
Overall cancers						
0	259,650	12,979	1,319,346.67	9.84	1(Ref.)	<0.0001
1	334,595	21,185	1,711,059.87	12.38	1.26 (1.23–1.29)	
2	328,186	23,003	1,656,449.03	13.89	1.41 (1.38–1.44)	
3	285,443	19,953	1,420,455.64	14.05	1.43 (1.39–1.46)	
4	207,646	14,238	1,012,356.88	14.06	1.43 (1.40–1.47)	
5	89,360	6437	426,614.01	15.09	1.54 (1.50–1.59)	
Liver						<0.0001
0	259,650	4018	1,345,045.29	2.99	1(Ref.)	
1	334,595	7392	2,345,505.60	3.15	1.06 (1.02–1.10)	
2	328,186	7846	2,680,088.93	2.93	1.00 (0.94–1.02)	
3	285,443	5581	2,521,278.33	2.21	0.74 (0.71–0.77)	
4	207,646	3175	1,960,972.12	1.62	0.54 (0.52–0.57)	
5	89,360	1400	920,740.46	1.52	0.51 (0.48–0.54)	
Biliary						<0.0001
0	259,650	250	1,354,435.46	0.18	1(Ref.)	
1	334,595	545	1,765,727.59	0.31	1.67 (1.44–1.94)	
2	328,186	694	1,713,856.28	0.40	2.19 (1.90–2.53)	
3	285,443	606	1,469,120.76	0.41	2.24 (1.94–2.60)	
4	207,646	475	1,046,063.01	0.45	2.48 (2.13–2.89)	
5	89,360	231	441,763.87	0.52	2.87 (2.40–3.43)	
Bladder						<0.0001
0	259,650	172	1,354,382.24	0.13	1(Ref.)	
1	334,595	386	1,765,390	0.22	1.72 (1.44–2.06)	
2	328,186	523	1,713,593.9	0.31	2.40 (2.02–2.85)	
3	285,443	494	1,468,657.14	0.34	2.66 (2.23–3.16)	
4	207,646	425	1,045,757.91	0.41	3.22 (2.70–3.85)	
5	89,360	199	441,566.99	0.45	3.59 (2.93–4.40)	
Esophagus						<0.0001
0	259,650	135	1,354,496.91	0.10	1(Ref.)	
1	334,595	197	1,765,958.15	0.11	1.11 (0.91–1.37)	
2	328,186	220	1,714,148.16	0.13	1.29 (1.05–1.57)	
3	285,443	194	1,469,416.35	0.13	1.32 (1.07–1.63)	
4	207,646	193	1,046,368.28	0.15	1.50 (1.19–1.85)	
5	89,360	66	441,912.08	0.15	1.50 (1.11–2.01)	
Laryngeal						<0.0001
0	259,650	52	1,354,694.77	0.04	1(Ref.)	
1	334,595	140	1,766,135.1	0.08	2.06 (1.50–2.84)	
2	328,186	177	1714418.44	0.10	2.69 (1.97–3.66)	
3	285,443	162	1,469,549.65	0.11	2.79 (1.88–4.13)	
4	207,646	112	1,046,543.83	0.11	2.80 (2.01–3.89)	
5	89,360	53	441,937.8	0.12	3.09 (2.27–4.21)	
Leukemia						<0.0001
0	259,650	129	1,354,546.93	0.1	1(Ref.)	
1	334,595	189	1,766,169.07	0.11	1.12 (0.90–1.40)	
2	328,186	212	1,714,518.59	0.12	1.30 (1.04–1.62)	
3	285,443	221	1,469,627.1	0.15	1.59 (1.28–1.97)	
4	207,646	166	1,046,523.58	0.16	1.68 (1.34–2.12)	
5	89,360	75	441,976.02	0.17	1.80 (1.50–2.13)	
Lung						<0.0001
0	259,650	963	1,353,109.74	0.71	1(Ref.)	
1	334,595	1816	1,763,511.24	1.03	1.44 (1.34–1.56)	
2	328,186	2158	1,711,355.12	1.26	1.77 (1.64–1.91)	
3	285,443	2007	1,466,855.44	1.37	1.93 (1.79–2.08)	
4	207,646	1562	1,044,462.24	1.50	2.12 (1.96–2.30)	
5	89,360	691	441,013.88	1.57	2.23 (2.02–2.46)	
Lymphoma						<0.0001
0	259,650	268	1,354,169.19	0.20	1(Ref.)	
1	334,595	413	1,765,390.75	0.23	1.18 (1.01–1.38)	
2	328,186	477	1,713,784.81	0.28	1.38 (1.17–1.63)	
3	285,443	411	1,469,064.72	0.28	1.41 (1.21–1.63)	
4	207,646	301	1,046,276.88	0.29	1.42 (1.21–1.65)	
5	89,360	131	441,755.53	0.29	1.51 (1.22–1.86)	
Multiple myeloma						<0.0001
0	259,650	73	1,354,665.95	0.05	1(Ref.)	
1	334,595	110	1,766,257.7	0.06	1.15 (0.86–1.55)	
2	328,186	155	1,714,582.95	0.09	1.68 (1.27–2.21)	
3	285,443	140	1,469,721.72	0.10	1.77 (1.33–2.35)	
4	207,646	105	1,046,641.76	0.10	1.86 (1.42–2.52)	
5	89,360	55	441,975.23	0.12	2.33 (1.64–3.31)	
Nerves						<0.0001
0	259,650	114	1,354,590.62	0.08	1(Ref.)	
1	334,595	214	1,766,091.27	0.12	1.44 (1.15–1.80)	
2	328,186	231	1,714,539.85	0.13	1.60 (1.28–2.00)	
3	285,443	168	1,469,699.8	0.14	1.68 (1.24–2.29)	
4	207,646	151	1,046,570.14	0.14	1.73 (1.35–2.20)	
5	89,360	66	441,961.96	0.15	1.78 (1.42–2.23)	
Oral						<0.0001
0	259,650	157	1,354,488.07	0.12	1(Ref.)	
1	334,595	320	1,765,725.54	0.18	1.57 (1.20–2.10)	
2	328,186	318	1,714,009.71	0.19	1.60 (1.30–1.90)	
3	285,443	295	1,469,253.33	0.20	1.74 (1.41–2.14)	
4	207,646	220	1,046,308.21	0.21	1.81 (1.50–2.19)	
5	89,360	97	441,892.7	0.22	1.90 (1.57–2.30)	
Pancreatic						<0.0001
0	259,650	240	1,354,462.12	0.18	1(Ref.)	
1	334,595	521	1,765,820.13	0.30	1.66 (1.43–1.94)	
2	328,186	647	1,714,141.81	0.38	2.13 (1.84–2.47)	
3	285,443	668	1,469,134.67	0.45	2.58 (2.22–2.98)	
4	207,646	553	1,046,195.85	0.53	3.01 (2.59–3.50)	
5	89,360	247	441,812.98	0.56	3.19 (2.67–3.82)	
Renal						<0.0001
0	259,650	185	1,354,295.16	0.14	1(Ref.)	
1	334,595	345	1,765,540.94	0.20	1.43 (1.20–1.71)	
2	328,186	413	1,713,765.23	0.24	1.76 (1.48–2.10)	
3	285,443	460	1,468,691.95	0.31	2.30 (1.94–2.72)	
4	207,646	412	1,045,740.36	0.39	2.90 (2.44–3.44)	
5	89,360	200	441,542.37	0.45	3.34 (2.73–4.08)	
Skin						0.0056
0	259,650	37	1,354,723.35	0.03	1(Ref.)	
1	334,595	54	1,766,370.81	0.03	1.12 (0.73–1.70)	
2	328,186	70	1,714,708.36	0.04	1.52 (0.98–2.36)	
3	285,443	66	1,469,802.02	0.04	1.65 (1.11–2.43)	
4	207,646	52	1,046,697.71	0.05	1.83 (1.23–2.72)	
5	89,360	24	442,015.79	0.05	2.02 (1.21–3.38)	
Stomach						<0.0001
0	259,650	1176	1,351,380.7	0.87	1(Ref.)	
1	334,595	2056	1,760,563.77	1.17	1.34 (1.25–1.44)	
2	328,186	2406	1,707,972.12	1.41	1.62 (1.51–1.74)	
3	285,443	2411	1,463,287.18	1.65	1.90 (1.77–2.04)	
4	207,646	1815	1,041,833.59	1.74	2.01 (1.87–2.16)	
5	89,360	796	439,998.98	1.81	2.09 (1.91–2.29)	
Thyroid						0.0001
0	259,650	1566	1,348,965.02	1.16	1(Ref.)	
1	334,595	1914	1,420,102.54	1.35	1.16 (1.09–1.24)	
2	328,186	1760	1,207,075.11	1.46	1.26 (1.17–1.35)	
3	285,443	1503	980,091.68	1.53	1.32 (1.23–1.42)	
4	207,646	1025	668,392.53	1.53	1.32 (1.22–1.44)	
5	89,360	457	240,327.73	1.90	1.41 (1.27–1.57)	
Cervical						0.26
0	147,274	152	758,779.63	0.20	1(Ref.)	
1	149,487	171	787,486.11	0.22	1.09 (0.87–1.35)	
2	129,524	127	678,426.82	0.19	0.93 (0.73–1.19)	
3	113,076	114	584,045.39	0.20	0.98 (0.78–1.24)	
4	82,287	102	418,459.67	0.24	1.22 (0.95–1.57)	
5	36,545	46	182,169.80	0.25	1.26 (0.91–1.75)	
Ovarian						0.57
0	147,274	163	758,835.14	0.21	1(Ref.)	
1	149,487	157	787,573.26	0.20	0.93 (0.75–1.16)	
2	129,524	138	678,527.99	0.20	0.95 (0.76–1.19)	
3	113,076	118	584,111.02	0.20	0.94 (0.74–1.19)	
4	82,287	87	418,548.71	0.21	0.97 (0.75–1.26)	
5	36,545	49	182,202.54	0.27	1.26 (0.91–1.73)	
Corpus						0.45
0	147,274	168	758,785.66	0.22	1(Ref.)	
1	149,487	174	787,526.35	0.22	1.00 (0.81–1.23)	
2	129,524	147	678,420.86	0.22	1.00 (0.78–1.22)	
3	113,076	133	584,041.72	0.23	1.03 (0.82–1.29)	
4	82,287	99	418,475.40	0.24	1.07 (0.84–1.37)	
5	36,545	54	182,165.06	0.30	1.35 (0.99–1.83)	
Breast						<0.0001
0	147,274	1308	755,595.15	1.73	1(Ref.)	
1	149,487	1244	724,519.51	1.72	0.99 (0.92–1.07)	
2	129,524	1030	57,438.09	1.79	1.04 (0.95–1.13)	
3	113,076	834	460,595.34	1.81	1.05 (0.96–1.15)	
4	82,287	584	312,082.50	1.87	1.08 (0.97–1.20)	
5	36,545	299	131,851.65	2.27	1.31 (1.15–1.50)	
Prostate						<0.0001
0	112,376	456	594,275.88	0.76732	1(Ref.)	
1	185,108	1127	975,527.38	1.15527	1.50 (1.35–1.68)	
2	198,662	1328	1,032,590.71	1.28609	1.68 (1.51–1.87)	
3	172,367	1270	882,278.37	1.43945	1.90 (1.70–2.11)	
4	125,359	1025	625,361.68	1.63905	2.18 (1.95–2.43)	
5	52,815	490	258,477.04	1.89572	2.53 (2.23–2.87)	
Testicular						0.09
0	112,376	10	595,563.22	0.016791	1(Ref.)	
1	185,108	10	978,501.45	0.01022	0.61 (0.25–1.46)	
2	198,662	13	1,036,019.35	0.012548	0.75 (0.33–1.71)	
3	172,367	10	885,612.4	0.011292	0.68 (0.28–,1.62)	
4	125,359	11	628,052.8	0.017514	1.05 (0.45–2.47)	
5	52,815	9	259,706.9	0.034654	2.08 (0.85–5.12)	

MS, Metabolic syndrome; HR, hazard ratio.

## Data Availability

Not applicable.

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
