# Peer review of "Influence of Metabolic Syndrome on Cancer Risk in HBV Carriers: A Nationwide Population Based Study Using the National Health Insurance Service Database"

_jcm, 2021, doi:10.3390/jcm10112401_

Round 1

Reviewer 1 Report

This article is a large, population based study, investigating whether metabolic syndrome (MS) increases the cancer risk in individuals with hepatitis B virus infection (HBV). The methods and results are quite clearly described. This study provides the interesting observations that MS increases the risk of most cancer types in people with HBV, but not HCC.

Major concerns:

  1. There are a very small number of references in the manuscript but several presumably relevant studies that have not been included (some examples: org/10.1002/hep.31612; doi.org/10.1053/j.gastro.2008.03.073; doi.org/10.1016/j.kjms.2012.12.006). In particular, doi.org/10.1002/hep.31612 reaches a different conclusion with regards to HCC, using data from a similar population.
  2. It is mentioned that the purpose of the study is to investigate synergistic effects of HBV and MS on cancer development. However, to truly demonstrate synergism, the effect of MS alone would need to be included (e.g. HBV, MS, HBV+MS) in the analysis. I note that synergism is not claimed in the discussion 
  3. Figure 3 is not mentioned in the text but is a good illustration of the data. Either remove the figure or discuss in the text

Minor concerns:

  1. Reference number 22 is a duplicate of number 18 (I think). Reference 21 appears to be a duplicate of 19. Reference 11 is a duplicate of reference 3. References 21-23 are also incomplete (all missing the journal name). There are other inconsistencies in reference formatting – inconsistent use of abbreviations and doi.
  2. The column labels for Table 3 are not intuitively understandable. Column 2 should be labelled “Number of Patients” or similar
  3. Line 39 – associate should be associated
  4. Line 191 – deceased should be decreased

Author Response

Dear Editor-in-Chief of JCM:

Thank you for helpful comments on the manuscript titled ‘Influence of Metabolic Syndrome on Cancer Risk in HBV Carriers: A Nationwide Population Based Study Using the National Health Insurance Service Database’ (jcm-1217623). We have revised our manuscript as suggested by the reviewers and agree to the points the reviewers have indicated. They are as follows:

[Reviewer’s Comments]

Reviewer 1

Major concerns:

  1. There are a very small number of references in the manuscript but several resumably relevant studies that have not been included (some examples: org/10.1002/hep.31612; doi.org/10.1053/j.gastro.2008.03.073; doi.org/10.1016/j.kjms.2012.12.006). In particular, doi.org/10.1002/hep.31612 reaches a different conclusion with regards to HCC, using data from a similar population.

--> Thank you for providing us with additional important studies related to the current topic of our research. We have reviewed and included pertinent data from the suggested papers you have mentioned in the discussion section as shown below.

<doi.org/10.1002/hep.31612>

“Recently, a study on the association of metabolic risk factors with risks of cancer and all‐cause mortality in patients with chronic hepatitis B was published using a similar but different data set from NHIS database. In the study by Lee et al. [17], the risk of developing HCC and non-HCC cancers both increased in patients with MS, which was different from the result of our study that showed negative association of HCC with the presence of MS. There could be several reasons for this discrepancy. First, while we analyzed the lipid profile after subdividing it into TG and HDL-c, Lee et al. used only total serum cholesterol level. Second, the waist circumference was substituted for BMI as one of the components of MS in their study, whereas we adopted the classic definition of MS. Third, the average age of the patients was younger [46 (37-54), median (IQR)] compared to our study population (53.5±9.77, mean±SD), which could have limited the assessment of cancer development due comparatively younger study population. More importantly, the number of patients lost to follow-up was over 50% in their study which could have acted as a shortcoming and render the data less representative of the population of interest.”

<doi.org/10.1016/j.kjms.2012.12.006>

“Interestingly, the study by Chen et al. [24] demonstrated that the relationship between MS and the occurrence of HCC was negligible. This is based on the observation in the HBV endemic area that the link between MS and HCC via development of NASH was trivial compared to the predominant influence of HBV infection per se on HCC development. However, the impact of MS on HCC was also insignificant in the population without HBV infection in their study, thus not being able to demonstrate the widely accepted MS-NASH-HCC sequence.”

<doi.org/10.1053/j.gastro.2008.03.073>

“Different from the study by Chen et al., the result of another study from Taiwan was quite similar to our study [18]. In this study, the authors demonstrated that DM was the most important risk factor for developing HCC in HBV carriers, and that central obesity was responsible for 33% increase in the risk of HCC. It was also shown that hypertriglyceridemia reduced the risk of HCC by 32%, which is in accordance with the finding of our study.”

  1. It is mentioned that the purpose of the study is to investigate synergistic effects of HBV and MS on cancer development. However, to truly demonstrate synergism, the effect of MS alone would need to be included (e.g. HBV, MS, HBV+MS) in the analysis. I note that synergism is not claimed in the discussion 

--> Thank you for your helpful comment. We agree with your suggestion that in order to truly demonstrate synergism, it is necessary to include the effect of MS alone. However, the population of our study was those with HBV infection and in order to investigate the effect of MS alone on cancer development, another dataset would be necessary. Nevertheless, since it is generally accepted that MS increases cancer risk in the general population, we considered “increased risk” as synergism. We have replaced “increased risk” with “synergism” in the discussion section where appropriate.

  1. Figure 3 is not mentioned in the text but is a good illustration of the data. Either remove the figure or discuss in the text

--> Thank you for your helpful comment. We have mistakenly marked Fig.3 as Fig.2 in line 116 in the Results section, and thus have made corrections accordingly.

Minor concerns:

  1. Reference number 22 is a duplicate of number 18 (I think). Reference 21 appears to be a duplicate of 19. Reference 11 is a duplicate of reference 3. References 21-23 are also incomplete (all missing the journal name). There are other inconsistencies in reference formatting – inconsistent use of abbreviations and doi.

--> Thank you for helpful comment. As you have suggested, we have deleted the duplicate and updated references

  1. The column labels for Table 3 are not intuitively understandable. Column 2 should be labelled “Number of Patients” or similar

--> Thank you for helpful comment. As you have suggested, we have changed the label of column 2 as suggested.

  1. Line 39 – associate should be associated

--> Thank you for helpful comment. Correction has been made as suggested.

  1. Line 191 – deceased should be decreased

--> Thank you for helpful comment. Correction has been made as suggested.

We agree with the reviewers in all points and the corrections in an annotated version are the points the reviewers have indicated.

 Thank you and the reviewers again for considering our manuscript to be published in Journal of Clinical Medicine. We look forward to receiving your answer soon.

Sincerely,

Jong Jin Hyun, M.D., Ph.D.

Reviewer 2 Report

Choe et al studied whether metabolic syndrome may be associated with different types of cancer risk among Koreans with chronic HBV using the national HCC screening program. Authors showed that only HCC was negatively associated with metabolic syndrome while other cancer types showed the positive correlation. This results are mainly driven by the metabolic factors (TG and HDL level) since other metabolic risk factors showed increased HCC risk in the Table 2. There are several issues on this conclusion.

First,  Authors should show the demographic table including age, sex, each metabolic syndrome (BMI, lipid profile, waist circumference, glucose) with number/percentage, medication use (whether on anti-viral), baseline HBV DNA, and other HBV related labs.

Second, it would be interesting to know whether the missingness and lost of f/u were random. If those who excluded were different from included participants, this can bias the results.

Third, in the methods, authors should describe in more details how their cox models were built (what were the co-variates) and what were the icd codes for HCC and other cancer. And how reliable is this approach to capture HCC diagnosis from this data? Is there any prior publication to support this?

Author Response

Dear Editor-in-Chief of JCM:

Thank you for helpful comments on the manuscript titled ‘Influence of Metabolic Syndrome on Cancer Risk in HBV Carriers: A Nationwide Population Based Study Using the National Health Insurance Service Database’ (jcm-1217623). We have revised our manuscript as suggested by the reviewers and agree to the points the reviewers have indicated. They are as follows:

[Reviewer’s Comments]

Choe et al studied whether metabolic syndrome may be associated with different types of cancer risk among Koreans with chronic HBV using the national HCC screening program. Authors showed that only HCC was negatively associated with metabolic syndrome while other cancer types showed the positive correlation. This results are mainly driven by the metabolic factors (TG and HDL level) since other metabolic risk factors showed increased HCC risk in the Table 2. There are several issues on this conclusion.

First,  Authors should show the demographic table including age, sex, each metabolic syndrome (BMI, lipid profile, waist circumference, glucose) with number/percentage, medication use (whether on anti-viral), baseline HBV DNA, and other HBV related labs.

--> According to your recommendation, we have added a new Table which describes the characteristics of the study patients (including age, sex, body mass index, smoking status, alcohol consumption, physical activity, and metabolic factors). However, the status or activity of HBV infection and the usage of antiviral agents could not be analyzed due to inherent limitation of the NHIS data as mentioned in the Discussion section.

Second, it would be interesting to know whether the missingness and lost of f/u were random. If those who excluded were different from included participants, this can bias the results.

--> We have added a paragraph in discussion to help readers understand the current limitation for being a retrospective study and also the potential possibility of selection bias as you have mentioned. The following paragraph was added as one of the limitations of our study.

“First, our study used retrospectively collected data from the NHIS database, Therefore, it is inevitable that there are missed information which may result in selection bias. However, this dataset comes from a nationwide population-based claims database with long-term follow-up of a large number of patients, thus reducing the potential for sampling bias.”

Third, in the methods, authors should describe in more details how their cox models were built (what were the co-variates) and what were the icd codes for HCC and other cancer. And how reliable is this approach to capture HCC diagnosis from this data? Is there any prior publication to support this?  

1) As suggested, the following paragraph was added in the Method section to describe how cox models were built.

“The multivariable analyses were adjusted for factors including age, sex, body mass index, smoking status, alcohol consumption and physical activity,”

2) The following sentence has been added to the Methods section as recommended.

“Presence of HCC was identified by ICD‐10 code C22, and non‐HCC cancers were also identified using ICD‐10 codes as shown in Supporting Table S1.”

3)  Korea NHIS database has a high HCC registration rate (ICD-10 code C22, 96.5%) and highly accurate diagnoses and has previously been validated as a reliable resource for research. Below is the reference published in Asian Pac J Cancer P.

Seo, H.J.; Oh, I.H.; Yoon, S.J. A Comparison of the Cancer Incidence Rates between the National Cancer Registry and Insurance Claims Data in Korea. Asian Pac J Cancer P 2012, 13, 6163-6168, doi:10.7314/Apjcp.2012.13.12.6163.

We agree with the reviewers in all points and the corrections in an annotated version are the points the reviewers have indicated.

 Thank you and the reviewers again for considering our manuscript to be published in Journal of Clinical Medicine. We look forward to receiving your answer soon.

Sincerely,

Jong Jin Hyun, M.D., Ph.D.